# Hemibiotrophic *Phytophthora infestans* Modulates the Expression of SWEET Genes in Potato (*Solanum tuberosum* L.)

**DOI:** 10.3390/plants12193433

**Published:** 2023-09-29

**Authors:** Hemant B. Kardile, Suhas Gorakh Karkute, Clarissa Challam, Nirmal Kant Sharma, Rahul Mahadev Shelake, Prashant Govindrao Kawar, Virupaksh U. Patil, Rupesh Deshmukh, Vinay Bhardwaj, Kumar Nishant Chourasia, Srikar Duttasai Valluri

**Affiliations:** 1ICAR-Central Potato Research Institute, Shimla 171001, India; nirmal_bittu@yahoo.co.in (N.K.S.); veerubt@gmail.com (V.U.P.); vinaycpri@gmail.com (V.B.); 2Department of Crop and Soil Science, 109 Crop Science Building, Oregon State University, Corvallis, OR 97331, USA; 3ICAR-Indian Institute of Vegetable Research, Varanasi 221305, India; suhaskarkute@gmail.com; 4ICAR-Central Potato Research Institute, Regional Station, Shillong 793009, India; clarissachallam@gmail.com; 5Division of Applied Life Science (BK21 Four Program), Plant Molecular Biology and Biotechnology Research Center, Gyeongsang National University, Jinju 52828, Republic of Korea; rahultnau@gmail.com; 6ICAR-Directorate of Floricultural Research, Zed Corner, Mundhwa Manjri Road, Mundhwa, Pune 411036, India; prashant.kawar@icar.gov.in; 7Department of Biotechnology, Central University of Haryana, Mahendergarh 123031, India; rupesh0deshmukh@gmail.com; 8ICAR-Central Research Institute for Jute and Allied Fiber, Barrackpore 700120, India; 9Department of Electrical Engineering and Computer Science, Oregon State University, Corvallis, OR 97331, USA; valluris@oregonstate.edu

**Keywords:** disease resistance, host susceptibility, late blight, potato, SWEET gene family

## Abstract

Sugar Efflux transporters (SWEET) are involved in diverse biological processes of plants. Pathogens have exploited them for nutritional gain and subsequently promote disease progression. Recent studies have implied the involvement of potato *SWEET* genes in the most devastating late blight disease caused by *Phytophthora infestans*. Here, we identified and designated 37 putative *SWEET* genes as *StSWEET* in potato. We performed detailed in silico analysis, including gene structure, conserved domains, and phylogenetic relationship. Publicly available RNA-seq data was harnessed to retrieve the expression profiles of *SWEET* genes. The late blight-responsive SWEET genes were identified from the RNA-seq data and then validated using quantitative real-time PCR. The *SWEET* gene expression was studied along with the biotrophic (*SNE1*) and necrotrophic (*PiNPP1*) marker genes of *P. infestans*. Furthermore, we explored the co-localization of *P. infestans* resistance loci and *SWEET* genes. The results indicated that nine transporter genes were responsive to the *P*. *infestans* in potato. Among these, six transporters, namely *StSWEET10*, 12, 18, 27, 29, and 31, showed increased expression after *P. infestans* inoculation. Interestingly, the observed expression levels aligned with the life cycle of *P. infestans*, wherein expression of these genes remained upregulated during the biotrophic phase and decreased later on. In contrast, *StSWEET13*, 14, and 32 didn’t show upregulation in inoculated samples suggesting non-targeting by pathogens. This study underscores these transporters as prime *P. infestans* targets in potato late blight, pivotal in disease progression, and potential candidates for engineering blight-resistant potato genotypes.

## 1. Introduction

Sugars Will Eventually be Exported Transporters (SWEET), are a class of transporter proteins involved in the sucrose export from parenchyma cells into the apoplasm [1]. The diverse physiological functions of SWEETs, such as sugar efflux, apoplasmic phloem loading, plant-pathogen interaction, and reproductive tissue development, have been confirmed in numerous studies [2,3]. Studies critically investigating the role of *SWEETs* under the plant pathogen interactions have revealed that the SWEET transporters are often hijacked by plant pathogens, possibly to provide sugars to the pathogen for their nourishment. The underlying mechanism for hijacking the SWEETs studied in different crop species has found that the effectors from the pathogen have a great role. For instance, Transcription Activator–Like (TAL) effectors produced by *Xanthomonas oryzae* were found to directly induce the rice *SWEET11* and *SWEET14* genes expression in infected cells and probably cause leakage of sugars that are then used by the pathogens for infection [4,5]. Such pathogenicity mechanism has been widely observed with different pathosystems.

A notable example is cassava blight, where effectors from the *Xanthomonas axonopodis* specifically induce the *MeSWEET10a* to promote virulence [6]. A similar mechanism was observed in fungal diseases, where effectors secreted by fungal pathogens induce *SWEET* gene expression. The virulence factor, NEP from necrotrophic *Botrytis cinerea,* along with the oxidative burst in plant cells, triggered a strong up-regulation of *VvSWEET4* expression in grapes which in turn promoted the fungal growth by the effect of enhanced hexose efflux and/or plant cell death [7]. Another mechanism involving the SWEETs providing the tolerance to fungal disease was reported in sweet potato where the expression level of *IbSWEET10* was significantly up-regulated upon infection of *Fusarium oxysporum.* It is evident from these studies that the effector derived modulation of *SWEET* genes of the host plants are the key players in plant-pathogen interactions.

Recently, three proteins, namely KRBP1, PP1c isoforms, and NRL1, have been identified in potato as the targets for the effector molecules released by the late blight-causing pathogen, *Phytophthora infestans* [8,9]). The roles of these proteins in supporting enhanced susceptibility could be manifested in at least two ways. They could either act directly as endogenous negative regulators of immunity, as shown for MPK4, or provide increased nutrients or other metabolites that promote pathogen growth, as typified by the *SWEET* genes [2]. The role of SWEETs under late blight disease conditions in potato has not been investigated yet. Considering these things, the present study was conducted to identify the role of *SWEET* genes during *P. infestans* infection in potato. To date, 35 *SWEET* genes have been reported in potato [10,11], but our systematic genome-wide analysis has predicted a total of 37 putative *SWEET* homologs in the potato genome. SWEET proteins were characterized in silico for structural variation and evolutionary relationship. Publicly available potato transcriptomics data for late blight disease was explored to pinpoint the *SWEET* genes having a role in *P. infestans* infection and subsequently validated by quantitative real-time PCR (qRT-PCR) along with the biotrophic and necrotrophic marker genes from the *P. infestans*. The promoter elements of *SWEET* genes were cloned and sequenced to identify sequence-level variation in the resistant and susceptible potato cultivars. As SWEETs have diverse physiological functions, we have also studied their tissue-specific expression using the qRT-PCR. The present study thus, provides for the first time the deeper details of the potential mechanism of resistance and also identified the most potential target SWEET genes for developing resistance against late blight disease in potato.

## 2. Results

### 2.1. Distribution of SWEET Genes in Potato Genome and Their Co-Localization with Loci Governing Resistance to P. infestans

A total of 37 SWEET genes with confirmed *MtN3_slv* domain were identified in the potato genome and accordingly denoted as *StSWEET* (Table 1). The *StSWEET* genes were named as per the order of location on chromosomes. A total of 132 resistance (R) genes and seven late blight resistance QTLs were mapped to the respective chromosomal positions along with the *StSWEET* genes (Figure 1). A total of five clusters of *SWEET* genes, one each on chromosomes 1, 3 and 4 and two on chromosome 6 were observed. Among these, the cluster on chromosome 3 harbors highest 13 *StSWEET* genes. Apart from the gene clusters rest, all *SWEET* genes were found to be uniformly distributed on different chromosomes except chromosome number 7 and 10, which is devoid of *StSWEET* genes.

Mapping of these genes revealed the contrasting pattern of co-localization of *StSWEET* genes and resistance loci. Chromosome number 3 has the maximum (13) number of *StSWEETs* with only 7 resistance loci to *P. infestans*. Contrastingly, chromosome number 8 and 11 houses the maximum number of resistance loci with the least number of *StSWEETs*.

Predicted gene structure reveals that all *StSWEET* genes have introns (Figure 2). *StSWEET* contains eight exons, maximum in number, followed by seven exons in *StSWEET10*, 25, and 26. Twenty out of 37 *StSWEET* are having six exons whereas *StSWEET2* having only 2 exons, minimum among all the SWEET family members.

### 2.2. In Sillico Analysis of StSWEET Proteins

The amino acid sequences of the 37 StSWEET proteins were used for the conserved domain search, and phylogenetic analysis. The length of StSWEET proteins ranges from 128 (StSWEET32) to 357 (StSWEET17) amino acids with the highest molecular weight of 39.9 kDa and a minimum of 14.4 kDa, respectively (Appendix A). Conserved domain analysis revealed that SWEET protein consists of MtN3_slv/PQ-loop repeat domain as their signature domain, 33 SWEET proteins contains two MtN3_slv/PQ-loop repeat domain whereas only four SWEET proteins have single MtN3_slv/PQ-loop repeat domain. There are approximately 90 amino acids in these two MtN3/saliva domains, all at the same positions in the proteins.

Transmembrane domain search performed with the TMHMM server identified 7 TMDs in 23 SWEETs and 3 TMDs in four SWEETs, and TMDs in between 7 and 3 were identified in ten proteins (Appendix A). Based on TMDs, SWEETs were classified as the SWEET and Semi-SWEET. Proteins with 3 TMDs are classified as Semi-SWEET, whereas TMDs greater than 3 are all classified as SWEET. Accordingly, potato have 62% SWEET and 38% Semi-SWEET proteins.

Phylogenetic analysis of these proteins revealed three major groups of proteins showing their early and late divergence (Figure 3). Group I proteins are early proteins. Group II proteins are composed of duplicated proteins from Group I while Group III members are the most recently evolved proteins with less difference in their sequence.

Blast2GO results predicted the putative function of all the StSWEET proteins, showing that almost all the proteins are involved in carbohydrate transport. Exceptionally StSWEET1 is predicted to be involved in the leaf senescence and cellular response to osmotic stress (Table 1).

Earlier studies showing the role of SWEET in many physiological processes indicated that StSWEET might play an important role in modulating gene expression. So, to identify the StSWEET isoforms having the potential to modulate the gene expression, we have predicted the presence of leucine-rich nuclear export signals (NES). Prediction results show that 24 out of 37 StSWEETs have the leucine-rich nuclear export signals (Table 1).

### 2.3. Prediction of cis-Acting Elements in the Putative Promoter Elements of StSWEET Genes

Prediction results show that the 1Kb upstream sequence taken for analysis possesses the characteristic features of the eukaryotic promoter. Apart from the characteristic cis-acting elements, the upstream sequences also house the characteristic elements responsive to different biological processes. We have grouped these elements into abiotic, biotic, wound, hormone, and light-responsive elements. Many uncharacterized elements in the putative promoter were also observed, whose function is yet to be discovered. The detailed analysis reveals that almost all the genes have *cis*-acting motifs for either biotic and abiotic stress or combination of both (Figure 4). Among the abiotic stresses, we found that out of 37 StSWEET promoters, 23 have heat stress (*HSE*), 20 have anaerobic conditions (ARE), 18 have drought (MBS), 2 have low temperature (*LTR*), and single isoform StSWEET9 possesses the GC-motif for anoxic conditions.

Interestingly, *StSWEET17* possess six MBS elements in its promoter in combination with one LTR. Under biotic and wound responsive elements, fifteen StSWEET proteins possess the fungal responsive elements (BoxW1 and Box E). The 24 SWEETs predicted for the presence of the TC-rich repeats, which is responsible for the defense. Pathogen attack often leads to wounding and subsequent elicitation of the immune response; the cis-acting elements for this mechanism involving the WUN-motif, W-box, Box-s, and ELI-box3 were predicted in almost all the isoforms with different combinations indicating the role of these genes under the pathogen attack. Considering the diverse role in regulating many physiological processes, the promoter elements of SWEET genes harboring the response elements for the plant hormones were expected. We have found response elements for six hormones in 12 StSWEET isoforms for ABA (ABRE), one (StSWEET21) for auxin (AuxRR-Core), 10 for ethylene (ERE), 22 for GA (GARE, P-box, TATC-box), 15 for MeJA (TGACG-motif, CGTCA-motif) and 9 for salicylic acid (TCA-element). Out of 37 genes, seven isoforms lack the light-responsive elements in their promoter, indicating the light independent mode of their action.

### 2.4. Expression Profiling of StSWEET Genes under P. infestans Challenge Inoculation

RNA sequencing data from three potato clones with varied degrees of response to *P. infestans* infection revealed that out 37 *StSWEET* isoforms, 28 isoforms get induced under late blight disease conditions. A total of five SWEETs namely *StSWEET18*, 12, 14, 27, and 13 showed more pronounced expression in susceptible cultivar clones, namely Desiree. The expression of these genes except *StSWEET18* was reduced in the rest two resistant cultivars namely Sarpo Mira and SW92-1015, an advanced breeding line (Figure 5; Appendix A). The expression profile of these five genes and the four additional candidate SWEETs was validated using the qRT-PCR in the Indian potato varieties *K. Bahar* (susceptible). The expression of *StSWEET* genes was correlated with the expression of *P. infestans* biotrophic (*SNE1*) and necrotrophic (*PiNPP1.1*) marker genes (Figure 6). Expression analysis of these marker genes from challenge inoculated KB revealed that the level of *SNE1* transcript increases at 1 dpi (days post inoculation) and thereafter level of transcript decreases up to 3 dpi. The *P. infestans* generally shift to the necrotrophic phase after 3 dpi and interestingly, the level of *PiNPP1.1* transcript also increases and reaches to its maximum at 3 dpi indicating the start of the necrotrophic phase. It is also evident from the necrotic symptoms appeared on the leaves at 3 dpi in detached leaf assay (Figure 7). We found a significant difference (Kruskal-Wallis test *p* < 0.01) in the mean daily necrotic lesion area between 2 dpi and 3 dpi, suggesting the onset of necrotic phase after 2 dpi.

We investigated the role of SWEET transporters under the late blight conditions in potato. It has been found that out of the 9 tested transporter genes, six (10, 12, 18, 27, 29 and 31) show the increased expression on challenge inoculation of *P. infestans* whereas the rest three (13, 14, and 32) do not show upregulation on challenge inoculation (Figure 8). The results show that *StSWEET10* was upregulated at 1 dpi, and the level of transcript increases after that up to necrotrophic phase. Following a similar trend initially, *StSWEET12* shows upregulation, but suddenly the expression level drops down at 3 dpi. The rest four genes (18, 27, 29 and 31) were up regulated upon challenge inoculation and then level of expression start reducing parallel with the end of the biotrophic phase and initiation of the necrotrophic phase. Intestinally, three isoforms, namely 13, 14, and 32, do not show upregulation upon the challenge inoculation of *P. infestans,* indicating that these isoforms may not be hijacked by the pathogens for their nutritional gain. Our results show the same trend in which *SNE1* and *PiNPP1.1* are coordinately expressed during biotrophy and necrotrophy, respectively, as previously reported by Lee and Rose [12].

## 3. Discussion

The study of plant-pathogen interactions in potato late blight disease has become easy with sequence information of both host and pathogen. It has been found in several crops, SWEET has a potential role in plant-pathogen interaction. Being a vegetatively propagated crop, potato tubers are often housing inoculum of various plant pathogens, and its load is aggravated as crop generation advances. Here in this study, we have explored the SWEET gene family in potato and investigated its role under late blight disease conditions in susceptible cultivars.

The number of SWEET proteins found in the potato genome slightly vary with the addition of two more genes in a previous report [11]. Gene cluster of 13 *StSWEETs* were mapped on chromosome 3, whereas chromosome 10 lacks *StSWEET* genes. A similar kind of pattern is also reported in tomato; wherein chromosome 3 is enriched, and chromosomes 10 and 11 are devoid of SWEET genes [10], showing the evolutionary conservation of these genes among different species of Solanaceae. Contrasting the co-localization pattern of *StSWEETs* and resistance loci to *P. infestans* reveals the role of these transporters in pathogen nutrition. Upregulated genes under challenge inoculation were mapped to chromosome 3, which harbored the minimum number of resistance loci to *P. infestans*.

Interestingly, all the predicted genes contain the intron in their structure indicating the nuclear origin, as a gene from the organellar origin generally does not include the intronic regions in their gene structure. Lack of intron-less *SWEET* genes may be because of the role of alternative splicing mechanism or due to their ancient origin since recently evolved genes tend to be intron less [13].

The presence of conserved MtN3_slv/PQ-loop repeat domain as well as presence of transmembrane domain in all StSWEET isoforms, shows its potential role in membrane transport [14,15]. It is also evident from the predicted function of these transporters, wherein almost all the proteins are involved in carbohydrate transport. The major function of *SWEET* genes is sugar transportation, as evident from other plant species like *Arabidopsis* and rice. Using the optical glucose sensors, it has been shown that *AtSWEET8*, 4,5, 7, and 11 and rice *OsSWEET11* and *OsSWEET14* function as low-affinity glucose transporters in *Arabidopsis* and rice, respectively [1,2]. Interestingly, *StSWEET1* is predicted to be involved in the leaf senescence and cellular response to osmotic stress indicating the potential of SWEETs for providing tolerance to abiotic stresses in potato. A similar kind of response is reported in *Arabidopsis*, where *AtSWEET15* is induced by osmotic stress mainly, drought, salinity, and cold [16], and also encodes the protein, which will lead the tissue to senescence [2]. In barley and tomato, *SWEET* genes were induced under salinity and cold stress, respectively [17,18].

SWEETs were classified as SWEET and semi-SWEET based on their TMDs. Potato contains 62 and 38 percent of SWEET and Semi-Sweet, respectively. The semi-SWEETs are the bacterial homologs among the smallest known transporters containing 3 TMDs [19,20]. Semi-SWEETs are widely distributed among the prokaryotes, with few exceptions. Many species have been studied for the atomic structure of Semi-SWEETs, as reviewed by Chen et al. [21]. Phylogenetic analysis of these proteins revealed three major groups of proteins showing their early and late divergence. Group I proteins are early proteins. Group II proteins are composed of duplicated proteins from Group I, while Group III members are the most recently evolved proteins with less different in their sequence.

Sugars act as compatible osmolytes whose content is dynamically regulated under stress conditions. It is hypothesized that SWEET may govern these processes in association with stress-specific transcription factors. The systematic in silico characterization of 5′ upstream regulatory elements reveals the presence of various cis-acting elements. The predicted cis-acting elements may provide the binding sites for transcription factors. *StSWEET* harbors cis-acting elements for various biotic and abiotic stresses along with hormone response and light-responsive elements. Interestingly, all the isoforms are predicted to contain different combinations of cis-acting elements responsive to wounding and the subsequent elicitation of the immune response.

Virulence mechanism in plant-pathogen interaction can be carried out by efficiently high-jacking the nutrient distribution machinery of the host. Among the nutrients, sugar is vital for the growth and development of both host and pathogen. SWEET genes are often targeted by pathogens to bring out the virulence in disease development. Our study showed that *P. infestans* induces the expression of six transporters (*StSWEET10*, 12, 18, 27, 29, and 31), whereas the expression of three transporters viz. *StSWEET13*, 14, and 32 remain unaffected in potato. Thus, our results are corroborated by earlier reports in *Arabidopsis,* rice and cassava where different SWEET genes are induced in response to bacterial pathogens [2,6] Interestingly, in our study, the expression of upregulated genes was modulated as per the life cycle of *P. infestans.* Under the biotrophic phase, *P. infestans* establish an intricate link with their host for survival. While perpetuating in the host system, the nutrient demand of the pathogen is accomplished from the host system. The *P. infestans* generally remains in the biotrophic phase up to 3 dpi which is evident from our results. Our expression data showed that the biotrophic marker gene, *SNE1* was induced upon challenge inoculation of *P. infestans*. The expression of the concerned gene remains upregulated up to 3 dpi, and then it decreases. No morphological symptoms were observed up to 3 dpi which shows the biotrophic growth of pathogens. During this phase, the pathogen acquires nutrition from the host machinery by diverting the flow of sugars for pathogen nutrition. This may be achieved by transcription induction of specific transporters involved in the sugar transport, specifically SWEETs. In rice, effector molecule from *Xanthomonas oryzae* directly interact with the promoter elements of SWEETs and induces the expression of OsSWEET11, 13, and 14 [22]. Our result also showed that *P. infestans* induces the expression of six StSWEET transporters. The expression of these transporters remained upregulated during the biotrophic phase of *P. infestans* (up to 3 dpi). The expression of upregulated genes starts decreasing after 3 dpi, which can be explained by the onset of the necrotrophic phase.

Induction of marker gene, *piNPP1*.1, and subsequent decrease in expression of *SNE1* at 3 dpi shows the onset of necrotrophy. Morphologically, the necrotrophic phase was observed with appearance of necrotrophic lesions on the leaves of potato plant. The phase is characterized by induced cell death with reduced nutrient demand as the pathogen kills the host cells and feeds on the dead tissues [23]. Hence, during this phase, host nutrient machinery get entirely paralyzed by the pathogen. Thus, it is evident that the expression of genes involved in nutrient transfer will decrease. A similar kind of expression pattern is observed in our results where expression of the upregulated *StSWEET* genes starts decreasing along with the SNE1. Contrastingly, the expression of PiNPP1.1 start increasing and remains upregulated up to 3 dpi.

The results showed that these six transporters could be the target of *P. infestans* for late blight development in potato. The exact mechanism of hijacking the potato nutrient machinery by *P. infestans* needs to be investigated. It is hypothesized that the virulent strategy used by *P infestans* may slightly differ from the virulence strategies used by the *X. oryzae* in rice, *B. cinera*, and *P. syringae* in grapes. In rice, virulence is achieved by direct interaction of effector molecules with promoter elements of *OsSWEET11*, 13, and 14 [22] whereas in grapes, *VvSWEET4* is induced by reactive oxygen species (ROS) and virulence factors from the *B. cinera* [7]. We speculate that the virulence strategy used by *phytopthora* in potato differs from the strategies reported earlier. We speculate that the *P. infestans* brings out the virulence by targeting the susceptibility factors from potato. Recently, three proteins, KRBP1, PP1c isoforms, and NRL1, have been identified in potato as the targets for the effector molecules released by *P. infestans* [8,9]. It is speculated that these proteins, along with the effector molecules from *Phytopthora* may target the SWEET and thereby providing nutrients or other metabolites that promote virulence. This hypothesis is also supported by the fact that the putative promoter elements of these SWEETs harbor cis-acting elements for fungal response with the few uncharacterized elements. The mechanism of induction of *StSWEET* under *P. infestans* infection is still poorly understood. It will become clearer from the interaction studies between the susceptibility factors, effector proteins and the promoter elements of SWEET. These interaction studies will identify the potential effector molecules targeting potato susceptibility factors and the binding site of these molecules in promoter elements of *StSWEET*.

Interestingly, three isoforms, 13, 14, and 32, do not show upregulation upon the challenge inoculation of *P. infestans,* indicating that these isoforms may not be hijacked by the pathogens for their nutritional gain. Similar results were obtained in grape wines where infection by *P. viticola* and *E. necator* did not lead to strong upregulation *VvSWEETs* indicating that these pathogens could not target the *SWEETS* of grapes [7]. It will be interesting to study these isoforms to look at the possible reasons for not being targeted by the pathogens and provide a clue to bring favorable changes in the other isoforms, which is most likely attacked by the *P. infestans* in the potato.

## 4. Materials and Methods

### 4.1. Genome-Wide Identification and Phylogenetic Evaluation of SWEET Genes

To identify *SWEET* gene family members, we searched genes annotated as *SWEET* using the keyword “bidirectional sugar transporter” in the Plaza database (http://plaza.psb.ugent.be/, accessed on 5 January 2017). In addition, *SWEET* genes were also identified by using BLAST searches performed against the potato whole genome sequence annotations provided at the Potato Genome Sequencing Consortium (PGSC) database using the SWEET gene sequence from *Arabidopsis* and tomato as a query. Sequence information for unique top matches was retrieved and further analyzed. Apart from this, resistance genes to *P. infestans* were also mined from the PGSC database using key word search and were mapped to the respective chromosomes along with potato SWEET genes.

Information about the position on chromosomes, number of introns, exons, and amino acids were also obtained from the Plaza database. Clustal omega was used for homologous sequence alignment using default settings, and the result was used to construct an un-rooted phylogenetic tree. The maximum likelihood method (MLM) was applied to construct a phylogenetic tree in which poisson correction, pairwise deletion, and bootstrapping served as default values to appraise the reliability of the tree.

To perform MSA and phylogenetic analysis of the amino acid sequences of potato SWEET family proteins, Mega 5.2 (http://www.megasoftware.net/, accessed on 5 January 2017) software was used.

### 4.2. Gene Structure Analysis

A schematic diagram of the gene structure of SWEET genes consisting of intron/exon structure was constructed using the Gene Structure Display Server (http://gsds.cbi.pku.edu.cn/, accessed on 5 January 2017).

### 4.3. In Silico Structural Analysis of StSWEET Proteins

The conserved domain was identified by executing a domain search by Conserved Domains Database (http://www.ncbi.nlm.nih.gov/Structure/cdd/cdd.shtml, accessed on 5 January 2017) and pfam database (REF) (http://pfam.sanger.ac.uk/, accessed on 5 January 2017). Only significant domains found in protein sequence were considered as a valid domains. Transmembrane domains in protein sequences of the 37 members of the potato MtN3/saliva/SWEET gene family were identified using TMHMM Server v.2.0 (http://www.cbs.dtu.dk/services/TMHMM, accessed on 5 January 2017).

To get more information about the nature of the StSWEET protein, the grand average of hydropathy (GRAVY), PI, and molecular weight was predicted by the ProtParam tool available on Expert Protein Analysis System (ExPASy) proteomics server (http://www.expasy.ch/tools/protparam.html, accessed on 5 January 2017). The stability of proteins was determined based on the instability index predicted using the ProtParam (https://web.expasy.org/cgi-bin/protparam/protparam, accessed on 5 January 2017).

### 4.4. Sub-Cellular Localization, Prediction of Leucine-Rich Nuclear Export Signals (NES), and Gene Ontology (GO) Annotations of StSWEETs

The subcellular localization of SWEET proteins was predicted by Subcellular Localization Prediction of Eukaryotic Proteins (SubLocV1.0) (http://www.bioinfo.tsinghua.edu.cn/SubLoc/eu_predict.htm, accessed on 5 January 2017), SVM based server ESLpred (http://www.imtech.res.in/raghava/eslpred/submit.html, accessed on 5 January 2017), and ProtComp9.0 server (http://linux1.softberry.com/berry.phtml, accessed on 5 January 2017). Similarly, leucine-rich nuclear export signals (NES) were predicted in all 37 StSWEET proteins using the online NES prediction tool (http://www.cbs.dtu.dk/services/NetNES/, accessed on 5 January 2017). 

SWEET protein function in terms of their Gene Ontology (GO) was predicted by Blast2GO (https://www.blast2go.com/, accessed on 5 January 2017). Localization consensus was predicted based on highest percentage of expected accuracy results.

### 4.5. Identification of cis-Elements in Putative Promoter Elements

To recognize the cis-acting elements upstream of *SWEET* genes, 1Kb upstream sequences of SWEET gene ORFs (putative proximal promoters) were obtained from the Plaza database (http://plaza.psb.ugent.be/, accessed on 5 January 2017). The online PlantCARE database (bioinformatics.psb.ugent.be/webtools/plantcare/html/, accessed on 5 January 2017) was used to identify the conserved motifs in the putative promoters of these *SWEET* genes.

### 4.6. Co-Localization StSWEET and R genes to P infestans

The *StSWEET* and *R* genes providing resistance to *P. infestans* were co-localized on a chromosomal map based on their physical position on the 12 *S. tuberosum* genome assembly using MapChart (version2.2) (https://www.wur.nl/en/show/Mapchart.htm, accessed on 5 January 2017) program.

### 4.7. Expression Dynamics of StSWEET Genes under Late Blight Disease in Potato

The publicly available transcriptomics data for late blight susceptible cultivar Desiree, and resistant clones Sarpo Mira and SW92-1015 under LB conditions were obtained from the SRA database (ERX278912, https://www.ncbi.nlm.nih.gov/sra, accessed on 5 January 2017). The raw reads were mapped on the potato genome using the CLC workbench. The FPKM values for StSWEET genes were obtained by filtering the concerned genes from the mapped reads.

### 4.8. Detached Leaf Assay for P. infestans

Detached leaf assay was performed for the Kufri Bahar using 45 days grown potato plant. Fourth leaves from the top were plucked from each replicate. Five leaves per replicate were placed in plastic trays with perforated plastic separators. Zoospore suspension of *P. infestans* isolate belonging to the A2 mating type was prepared as per the methodology given by Shandil et al. [24]. The strain was previously isolated at our laboratory from late blight-infected potato fields in Shimla, placed in mid-Himalayas with wet temperate climate (31.61° N, 77.10° E) and has been submitted to gene bank for agriculturally important microbes at ICAR-Central Potato Research Institute, Shimla with No. HP10-31 [25]. The strain was cultured on tuber slices of cv. Kufri Jyoti. Zoospore suspension was prepared by washing the infected tuber surface in sterilized distilled water and diluted to 40,000 sporangia/mL. The sporangial suspension was incubated at 4 °C for 30 min to release zoospores. The leaves were inoculated with the 20 µL zoospore suspension using an autopipette. The inoculated leaves were incubated at 18 ± 1 °C for 5 days. High humidity was maintained using a moist foam sheet. Lesion area (cm^2^) from challenge inoculated leaf was calculated using sketchandcalc (https://www.sketchandcalc.com/, accessed on 5 January 2017). at a daily interval.

### 4.9. Plant Material and Validation of StSWEET Gene Expression Profiles along with the Biotrophic and Necrotrophic Markers of P. infestans by Quantitative Real-Time PCR

Kufri Bahar (KB), an LB susceptible Indian potato cultivar was used in this study. All the plant material was obtained from the Potato Germplasm lab, ICAR-Central Potato Research Institute, Shimla, India. Well-sprouted tubers of similar size were planted in the potting mixture in 15 cm pots with five replications per cultivar for each experiment. The complex strains of *P. infestans* were used for challenge inoculation. Leaf samples for RNA isolation were collected before and at 24 h intervals after inoculation up to 120 h. The functional Late blight responsive StSWEET genes shortlisted based on the digital expression profiles were validated using the qRT-PCR in KB along with the biotrophic marker, SNE1, and necrotrophic marker, PiNPP1 of *P. infestans*.

### 4.10. RNA Extraction and qRT-PCR

Total RNA was isolated using a NucleoSpin^®^ RNA kit (MACHEREY-NAGEL) following the user’s instruction. First-strand cDNAs were synthesized from RNase-free DNase I treated (Invitrogen, Waltham, MA, USA) total RNA to eliminate genomic DNA contamination using a High Capacity cDNA synthesis™ system (Applied Biosystems, Waltham, MA, USA). qRT-PCR was performed with ABI Prism 7900 HT Real-time PCR system (Applied Biosystems, Waltham, MA, USA) using Power SYBR^®^ Green PCR Master Mix (Applied Biosystems, Waltham, MA, USA). Three replicates were performed for the analysis of each sample. Potato 18S-rRNA as an internal control and the relative expression levels of the target gene were determined. For relative quantification, the 2^−ΔΔCT^ method between conditions in RT-qPCR was applied. The sequence of the primers is listed in Appendix A.

## 5. Conclusions

The study reveals the dynamic gene regulation of *StSWEET* under *P. infestans* attack in potato. This study has led to the identification of the SWEET as the putative susceptibility factor of late blight in potato. Our study has identified 9 transporter genes responsive to the *P. infestans* infection in potato. Out of 9 SWEETs, six genes, namely *StSWEET10*, 12, 18, 27, 29, and 31, show increased expression on challenge inoculation of *P. infestans,* whereas the rest three *StSWEET13*, 14, and 32 did not show the upregulation on challenge inoculation. The study reveals the role of these transporters as a major susceptibility factor in late blight disease conditions and can be manipulated to engineer the late blight tolerant potato genotypes.

## Figures and Tables

**Figure 1 plants-12-03433-f001:**
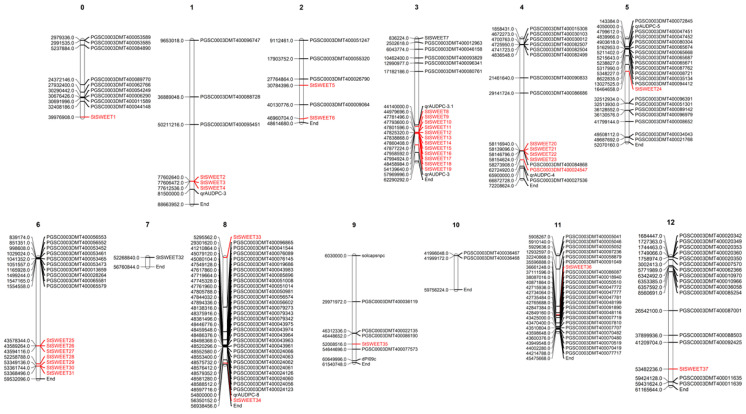
Genome-wide distribution of *StSWEET* and *R* genes to *P. infestans* on twelve chromosomes of potato.

**Figure 2 plants-12-03433-f002:**
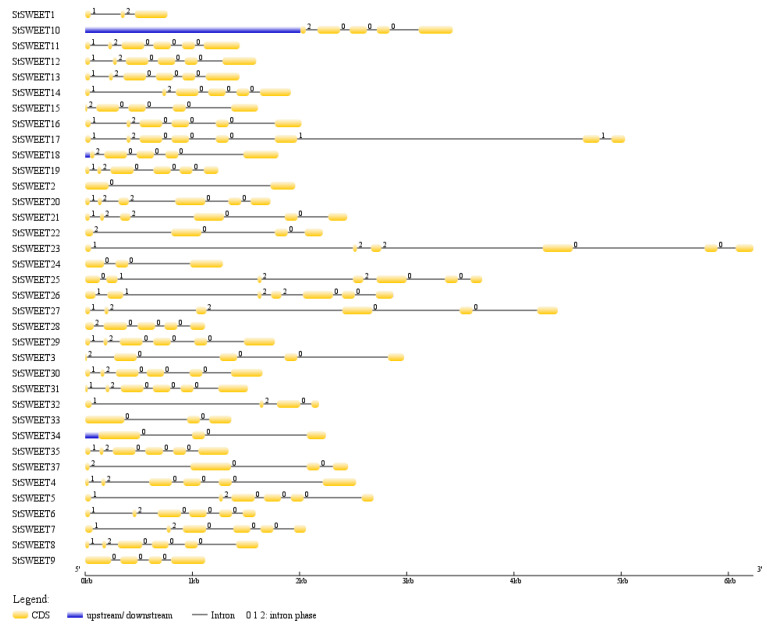
Gene structure showing the exon-intron boundaries in StSWEETs.

**Figure 3 plants-12-03433-f003:**
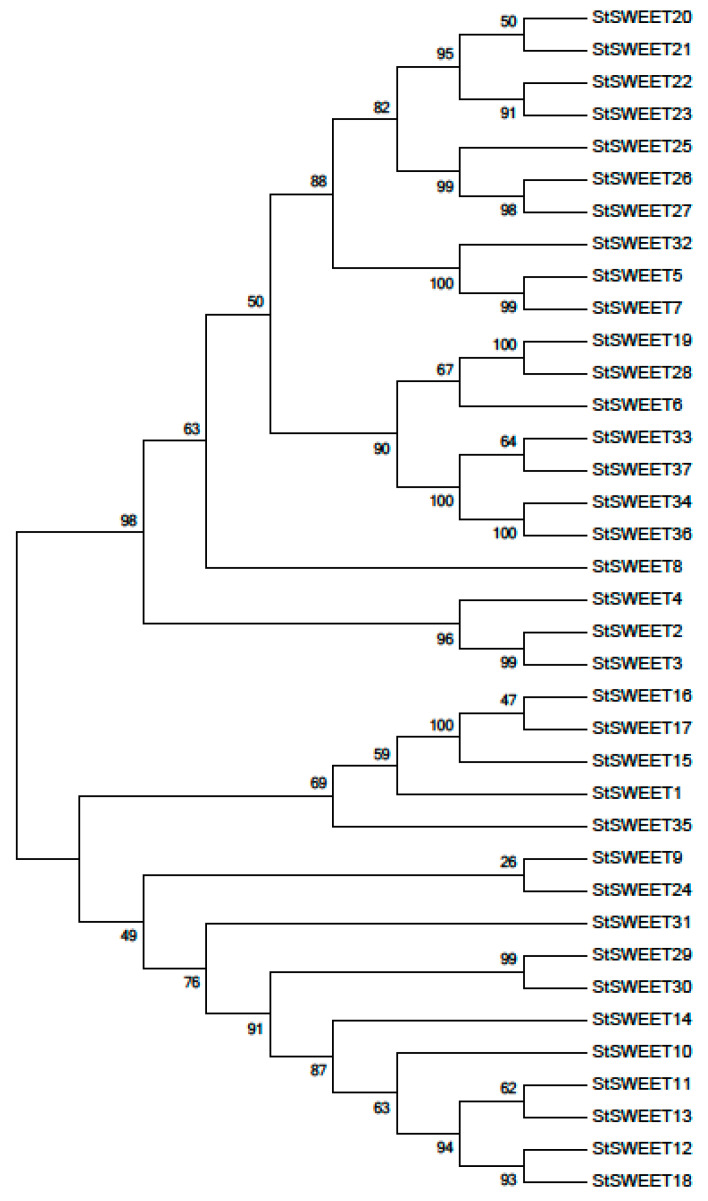
Phylogenetic distribution of SWEET genes identified in potato genome. The tree was constructed using MEGA 5.2 software tool.

**Figure 4 plants-12-03433-f004:**
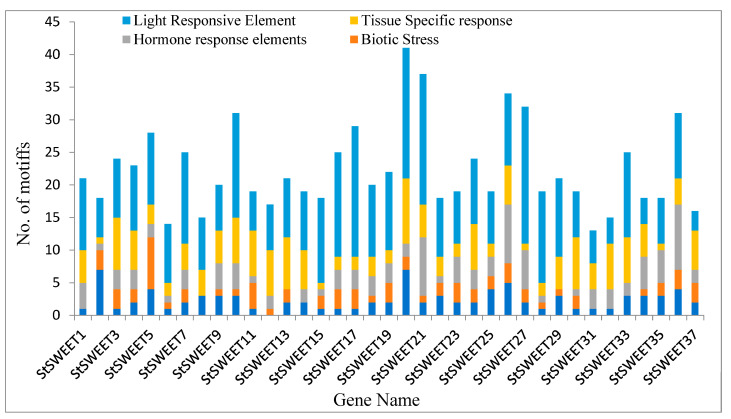
*cis*-acting motifs for all SWEET genes of potato.

**Figure 5 plants-12-03433-f005:**
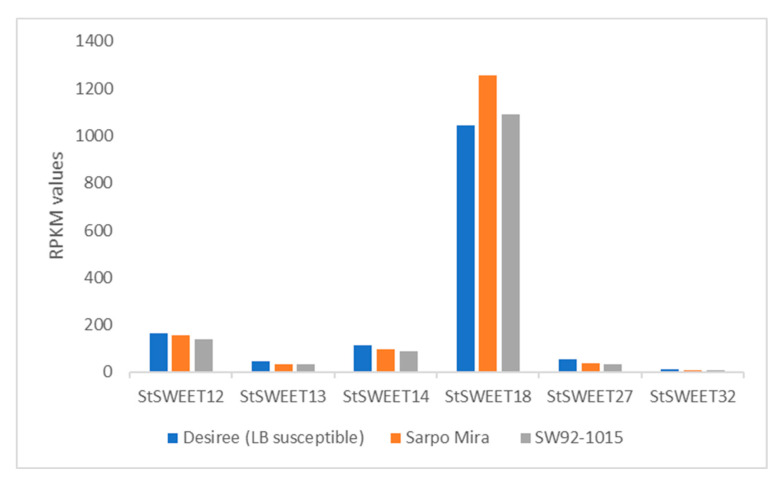
RPKM (Reads Per Kilobase Million) values of SWEET genes in late blight resistant and susceptible cultivars.

**Figure 6 plants-12-03433-f006:**
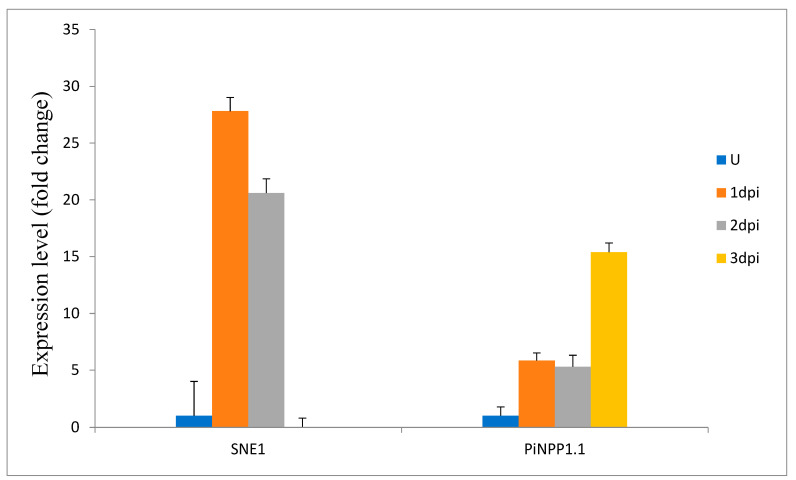
Expression analysis of *P. infestans* genes *SNE1* and *PiNPP1* using qRT PCR at different time interval (U: preinoculation, dpi: days post inoculation).

**Figure 7 plants-12-03433-f007:**
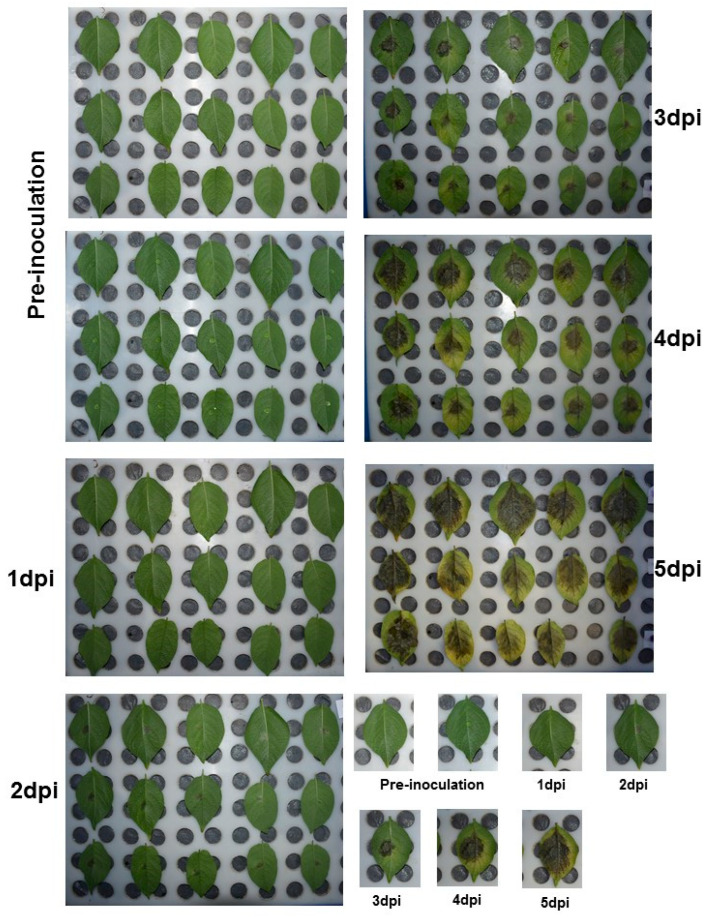
Detached leaf assay performed with the K. Bahar showing the necrotrophic lesions, asignificant difference (Kruskal-Wallis test *p* < 0.01) is found in the mean daily necrotic lesion area between 2 dpi and 3 dpi, suggesting the onset of necrotic phase after 2 dpi (days to post inoculation).

**Figure 8 plants-12-03433-f008:**
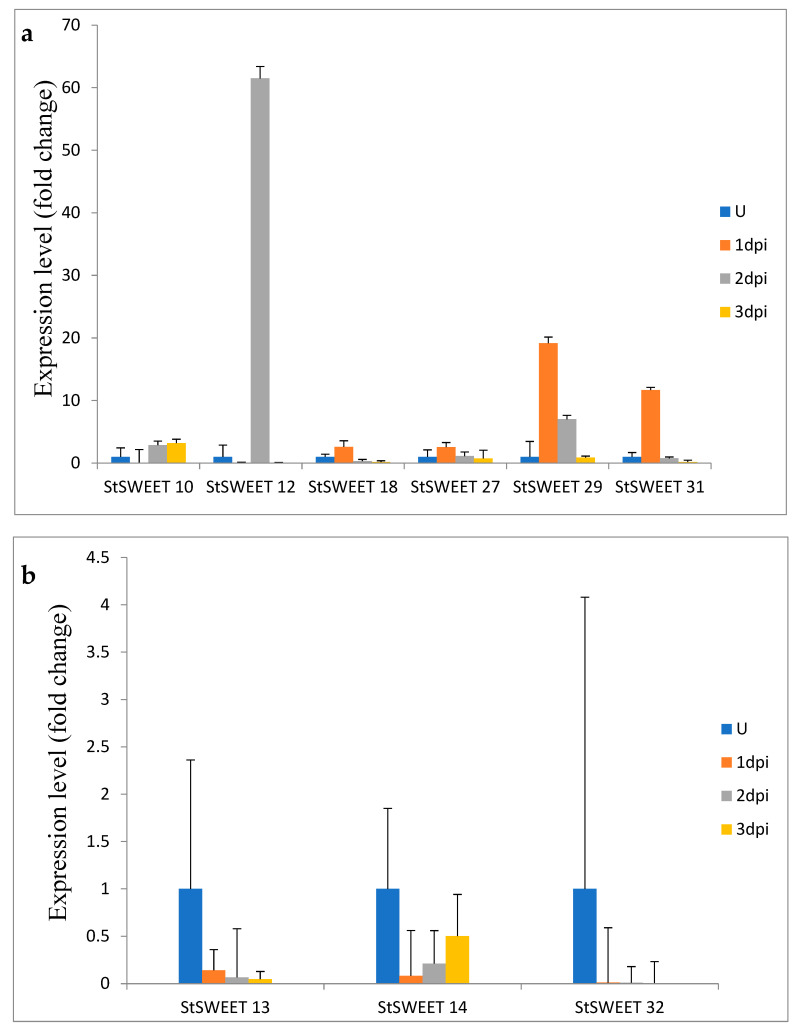
SWEET transporters under the late blight conditions in potato, (**a**) qRT PCR gene expression level; (**b**) qRT PCR gene expression level. (U: preinoculation, dpi: days post inoculation).

**Table 1 plants-12-03433-t001:** Details of transmembrane domains, conserved domains, subcellular localization, protein length, and biochemical properties predicted for 37 StSWEET genes identified in potato genome.

SWEET Name	Locus ID	Functional Annotation	Protein Length (Molecular Weight)	TM Domain	Sub-Cellular Localization	Hydro-Pathy	SWEET Semi-SWEET
StSWEET1	LOC102591902	BDST sweet12-like	14,823.5	3	Plasma membrane	0.666	Semi-SWEET
StSWEET2	Sotub01g037650	BDSTsweet17like	16,937.6	3	_	0.336	Semi-SWEET
StSWEET3	Sotub01g037660	BDSTsweet16like	24,470.8	5	_	0.597	Semi-SWEET
StSWEET4	Sotub01g037670	BDST sweet2alike	32,147.2	7	_	0.408	SWEET
StSWEET5	Sotub02g018000	BDST sweet6alike	25,571.5	7	Plasma membrane	1.022	SWEET
StSWEET6	Sotub02g035970	BDST sweet2alike	25,761.9	7	Plasma membrane	0.751	SWEET
StSWEET7	Sotub03g006170	BDST sweet3like	26,431.4	7	Plasma membrane	0.791	SWEET
StSWEET8	Sotub03g013630	BDST sweet12like	29,775.5	7	Plasma membrane	0.418	SWEET
StSWEET9	Sotub03g018270	BDST sweet12like	32,153.5	6	Plasma membrane	0.552	Semi-SWEET
StSWEET10	Sotub03g018280	BDST sweet12like	32,782.3	7	Plasma membrane	0.668	SWEET
StSWEET11	Sotub03g018290	BDST sweet12like	33,762.3	7	Plasma membrane	0.661	SWEET
StSWEET12	Sotub03g018300	BDST sweet12like	33,086.8	7	Plasma membrane	0.673	SWEET
StSWEET13	Sotub03g018310	BDST sweet12like	33,533.2	7	Plasma membrane	0.646	SWEET
StSWEET14	Sotub03g018320	BDST sweet14like	32,061.6	7	Plasma membrane	0.734	SWEET
StSWEET15	Sotub03g018330	BDST sweet12like	28,119.5	6	Plasma membrane	0.567	Semi-SWEET
StSWEET16	LOC102603273	BDST sweet14-like	30,780.5	7	Plasma membrane	0.569	SWEET
StSWEET17	LOC102603614	BDST sweet14-like	39,953	7	Plasma membrane	0.401	SWEET
StSWEET18	Sotub03g022530	BDST sweet5like	32,415.1	6	Plasma membrane	0.615	Semi-SWEET
StSWEET19	Sotub03g027590	BDST sweet1like	26,664.8	7	_	0.641	SWEET
StSWEET20	Sotub04g024600	BDST n3like	27,791	7	Plasma membrane	0.658	SWEET
StSWEET21	Sotub04g024590	BDST sweet1like	27,669	7	Plasma membrane	0.604	SWEET
StSWEET22	Sotub04g024580	BDST sweet1like	24,431.5	6	_	0.551	Semi-SWEET
StSWEET23	Sotub04g024570	BDST sweet1like	27,967.5	7	_	0.746	SWEET
StSWEET24	Sotub05g020570	BDST sweet12like	22,166.2	4	Plasma membrane	0.621	Semi-SWEET
StSWEET25	LOC102600141	BDST sweet1-like	33,207.1	7	_	0.483	SWEET
StSWEET26	LOC102599475	BDST sweet1-like	35,193.5	6	_	0.515	Semi-SWEET
StSWEET27	LOC102599149	BDST sweet1-like	28,236.5	7	_	0.706	SWEET
StSWEET28	Sotub06g029710	BDST sweet2like	26,672.8	7	_	0.739	SWEET
StSWEET29	Sotub06g028350	BDST sweet5like	32,750.1	7	Plasma membrane	0.646	SWEET
StSWEET30	Sotub06g028340	BDST sweet12like	33,158.6	7	Plasma membrane	0.727	SWEET
StSWEET31	Sotub06g028330	BDST sweet12like	31,290.5	7	Plasma membrane	0.708	SWEET
StSWEET32	Sotub07g025190	BDST sweet7like	14,442.9	3	_	0.401	Semi-SWEET
StSWEET33	LOC102603968	BDST sweet4-like	26,330.8	6	_	0.649	Semi-SWEET
StSWEET34	Sotub08g027480	BDST sweet14like	25,698.8	6	Plasma membrane	0.607	Semi-SWEET
StSWEET35	Sotub09g022640	BDST sweet7like	32,627.5	7	_	0.451	SWEET
StSWEET36	Sotub11g020620	BDSTsweet4like	15,062.2	3	_	0.983	Semi-SWEET
StSWEET37	Sotub12g024450	BDST sweet17like	25,321.5	6	_	0.709	Semi-SWEET

bidirectional sugar transporter-BDST.

## Data Availability

Not applicable.

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
