# Peer review of "Hemibiotrophic Phytophthora infestans Modulates the Expression of SWEET Genes in Potato (Solanum tuberosum L.)"

_plants, 2023, doi:10.3390/plants12193433_

Round 1
Reviewer 1 Report
The main goal of the paper titled “Hemibiotrophic Phytophthora infestans modulates the expression of SWEET genes in potato (Solanum tuberosum L.)” was to study and reveal the dynamic gene regulation of StSWEET under P. infestans attack in potato. The study led to the identification of the SWEET as the putative susceptibility factor of late blight in potato. The authors identified 9 SWEET genes and 6 of them increased expression on challenge inoculation of P. infestans. The SWEET genes function as susceptibility genes is not really a novelty however the authors identified more genes and went into deeper detail regarding the potential mechanisms.
In this perspective, the main drawback of the manuscript has some novelty that can be useful, especially for practical applications. However, the manuscript should be revised for publication as some aspects need clarification.
Introduction
The introduction section should be revised to make it more concise. In this form, it contains a lot of information irrelevant to the current study. Also, the authors are advised to highlight the novelty of their work at the end of this section.
The authors are encouraged to make a statistical analysis of the resistance of the potato varieties selected.
Materials and method
This section should be divided into subsections according to the author's guidelines.
The method used for the antifungal assay should be seriously revised as in this form is totally unclear. The major problem is in this section as NO statistical analysis seems to be done regarding the lesions caused by Phytophthora infestans on the different varieties compared. Please provide statistical analysis. Moreover, several other information are necessary. Which strain of P. infestant was used? The strain was isolated by the authors? If so, how it was identified? Where is it deposited? Why did the authors not use a standard strain for the assay?
In addition, it is necessary to report in this section the characteristics of the potato varieties used. Desiree, Sarpo Mira and SW93-1015 are reported in the results but no mention is made in Materials and methods.
Results
Some parameters need to be defined just after appearing in the text
Legends of the figures should be completed indicating for example the meaning of U, and dpi (see for example in Figure 6.)
Discussion
All the findings of the current work need to be compared and discussed with the results of other researchers finding instead of having a general comparison with other researchers' works. The authors should perform a comparison between the forecasting results. In your discussion section, please link your empirical results with a broader and deeper literature review.
Author Response
Authors are very much thankful to the reviewer for sparing valuable time and giving critical comments to improve the overall manuscript.
We have improved the manuscript as per your comments.
Introduction
The introduction section should be revised to make it more concise. In this form, it contains a lot of information irrelevant to the current study. Also, the authors are advised to highlight the novelty of their work at the end of this section.
The authors are encouraged to make a statistical analysis of the resistance of the potato varieties selected.
Reply: Thank you for your critical comments to improve the overall manuscript. The introduction section has been revised to make it more concise as per the suggestion. The novelty has been highlighted at the end of the introduction.
Materials and method
This section should be divided into subsections according to the author's guidelines.
The method used for the antifungal assay should be seriously revised as in this form is totally unclear. The major problem is in this section as NO statistical analysis seems to be done regarding the lesions caused by Phytophthora infestans on the different varieties compared. Please provide statistical analysis. Moreover, several other information are necessary. Which strain of P. infestant was used? The strain was isolated by the authors? If so, how it was identified? Where is it deposited? Why did the authors not use a standard strain for the assay?
Reply: Thanks for suggesting to perform the statistical analysis to study the biotrophic and necrotrophic phase of pathogen. We performed the Kruskal-Wallis test to provide the statistical evidence for onset of necrotrophic phase after 2dpi. Following line has been added to results section 2.4:
“We found a significant difference (Kruskal-Wallis test p<0.01) in the mean daily necrotic lesion area between 2 dpi and 3 dpi, suggesting the onset of necrotic phase after 2dpi.”
We have improved the materials and methodology section (4.8) by adding following lines:
“Necrotic lesions area was calculated using sketchandcalc (https://www.sketchandcalc.com/).”
Thanks for your questions on P. infestans stain details. We have used strain of P. infestans belonging to A2 mating type previously isolated by one of author from late blight-infected potato fields in Shimla, placed in mid-Himalayas with wet temperate climate (31.61°N,77.10°E). The specimen P. infestans A2 Mating Type has been submitted to gene bank for agriculturally important microbes at ICAR-Central Potato Research Institute, Shimla with No. HP10-31 (Patil et al., 2017). Since these stains are more virulent with K. Bahar (species under investigation), so we chose to use these stains for further investigation. Strains used for this study is mentioned on line 424.
In addition, it is necessary to report in this section the characteristics of the potato varieties used. Desiree, Sarpo Mira and SW93-1015 are reported in the results but no mention is made in Materials and methods.
Reply: We have used the publicly available transcriptome data of late blight susceptible cultivar Desiree, and resistant clones Sarpo Mira and SW93-1015 under LB conditions to mine the expression profiles of SWEET genes under late blight. This information is added to the materials and methods section, 4.7
Results
Some parameters need to be defined just after appearing in the text
Legends of the figures should be completed indicating for example the meaning of U, and dpi (see for example in Figure 6.)
Reply: Thanks for the suggestion. We have changed text in the figure legends and results section.
Thanks for pointing out this. We have changed the figures legends and added the meaning of U, and dpi in Figure 6.)
Discussion
All the findings of the current work need to be compared and discussed with the results of other researchers finding instead of having a general comparison with other researchers' works. The authors should perform a comparison between the forecasting results. In your discussion section, please link your empirical results with a broader and deeper literature review.
Reply: Thanks for suggestion to improve the discussion section. We have improved the discussion section by corelating our findings with recent literature.
Reviewer 2 Report
The work describes the results of study on the effect of Phytophthora infestans on the expression of SWEET genes in potato. The subject is interesting and the results provide an advancement of the current knowledge of transporter genes responsive to the P. infestans infection in potato. This paper is suitability for publication in Plants. The methods are adequately described. The results are clearly presented and the conclusions are supported by the results. Overall the well written manuscript can be accepted for publication in Plants after revision have been made.
Manuscript should be prepared according to the Instructions for Authors. In the text, reference numbers should be placed in square brackets [ ].
Keywords should reflected the scientific content of the work but should not be repetition of the title words (potato, phytophthora, SWEET). Please find such words (which are not be repetition of the title) that more detail reflecting the scientific content of the work.
The hypothesis made in the study and the main aim of the work should be clearly stated.
Please check the Figure numbers in the text (line 137 – Fig. 4a, line 140 – Fig. 3, line 165 – Fig. 4).
Figure 8: Please added “a” and “b”.
Line 71: “several other studies” – please added references.
Line 130: “(Supplementary file 2)” – there should be number of supplementary table.
Line 155: “(Supplementary file 3)” – there should be number of supplementary table.
Author Response
Authors are very much thankful to the reviewer for sparing the valuable time to critically review the manuscript and giving comments to improve it.
We have improved the manuscript as per your suggestions.
Manuscript should be prepared according to the Instructions for Authors. In the text, reference numbers should be placed in square brackets [ ].
Reply: The manuscript has been prepared in the format of plants by placing the numbers for reference in square brackets.
Keywords should reflected the scientific content of the work but should not be repetition of the title words (potato, phytophthora, SWEET). Please find such words (which are not be repetition of the title) that more detail reflecting the scientific content of the work.
Reply: Thank you for the suggestion. The keywords have changed as per your suggestion.
The hypothesis made in the study and the main aim of the work should be clearly stated.
Reply: The introduction part is revised to make it more concise and reflect the aim of the work.
Please check the Figure numbers in the text (line 137 – Fig. 4a, line 140 – Fig. 3, line 165 – Fig. 4).
Reply: Thank your noticing the errors. We have made necessary correction for these comments.
Figure 8: Please added “a” and “b”.
Reply: The a and b names have been added in the figure 8 now.
Line 71: “several other studies” – please added references.
Reply: It has been modified as per the suggestion.
Line 130: “(Supplementary file 2)” – there should be number of supplementary table.
Reply: Corrected as per the suggestion. It should be table 1. Thank you for noticing.
Line 155: “(Supplementary file 3)” – there should be number of supplementary table.
Reply: Corrected as per the suggestion. It should be table 1. Thank you for noticing.
Round 2
Reviewer 1 Report
Dear authors,
thank you for improving the manuscript.